# Effect of High-Pressure Processing on the Qualities of Carrot Juice during Cold Storage

**DOI:** 10.3390/foods12163107

**Published:** 2023-08-18

**Authors:** Chiu-Chu Hwang, Hung-I Chien, Yi-Chen Lee, Chung-Saint Lin, Yun-Ting Hsiao, Chia-Hung Kuo, Feng-Lin Yen, Yung-Hsiang Tsai

**Affiliations:** 1Department of Seafood Science, National Kaohsiung University of Science and Technology, Kaohsiung 811213, Taiwan; omics1@gmail.com (C.-C.H.); aaoc1018@yahoo.com.tw (H.-I.C.); lionlee@nkust.edu.tw (Y.-C.L.); cdd828@gmail.com (Y.-T.H.); kuoch@nkust.edu.tw (C.-H.K.); 2Department of Food Science, Yuanpei University of Medical Technology, Hsinchu 300150, Taiwan; chungsl@mail.ypu.edu.tw; 3Department of Fragrance and Cosmetic Science, Kaohsiung Medical University, Kaohsiung 807378, Taiwan; flyen@kmu.edu.tw

**Keywords:** high hydrostatic pressure, carrot juice, quality, blanching, shelf life

## Abstract

This study examines the impact of blanching (heating at 85 °C for 60 s), high-pressure processing (HPP) (600 MPa, 3 min, 20 °C), and a combination of both blanching and HPP on the microbiological and chemical qualities, colour, and antioxidant properties of carrot juice stored at 4 °C for 15 days. In terms of microbiological quality, the total plate count (TPC), coliform bacteria, and *Salmonella* spp. rose rapidly in the control group (untreated) as the storage time increased. However, for the blanching group, these values climbed more gradually, surpassing the microbiological limits for juice beverages (TPC < 4 log CFU/mL, Coliform < 10 MPN/mL, and *Salmonella* spp. negative) on the 9 days of storage. In contrast, TPC, coliforms, and *Salmonella* spp. were undetectable in the HPP and blanching/HPP samples throughout the storage period. Additionally, as storage time lengthened, the pH, total soluble solids, and Hunter colour values (*L*, *a*, *b*) diminished in the control and blanching groups, whilst titratable acidity and browning degree intensified. However, the HPP and blanching/HPP noticeably delayed these decreases or increases. Moreover, although the total phenolic content and DPPH radical scavenging ability in the HPP samples remained relatively stable during storage and were lower compared to other groups, the β-carotene content was higher at the end of the storage period. In summary, HPP can effectively deactivate microorganisms in carrot juice, irrespective of whether blanching is applied, and can impede reductions in pH, increases in acidity, and colour changes, ultimately extending the juice’s shelf life.

## 1. Introduction

Carrots are a bountiful source of plant nutrients, including antioxidants, β-carotene, minerals, and vitamins [1]. Carrot juice, one of the main products derived from carrots, boasts a plethora of health benefits. However, as fresh and unprocessed carrot juice is classified as a low-acid food with a pH of around 6.0, it is susceptible to microbial growth, making it prone to spoilage with a limited shelf life and posing potential food poisoning risks [2]. Thus, commercial carrot juice production employs techniques such as thermal sterilization, blanching, and acidification to reduce the number of microbes and prolong shelf life [3]. Nonetheless, high-temperature treatments between 105 °C and 121 °C can adversely affect the heat-sensitive nutrients, texture, colour, and flavour present in carrot juice [4].

Blanching is primarily used to deactivate naturally occurring enzymes in plant-based foods. This process typically involves temperatures of 70–85 °C, lasting for several seconds up to tens of seconds [5]. In fruit and vegetable processing, blanching is frequently employed to disable enzymes, thereby preventing enzymatic browning [6]. Therefore, blanching has become an essential pretreatment step in vegetable juices, such as carrot juice, where it helps to maintain colour, neutralize enzymes and microbes, and eliminate trapped air [5]. Studies have shown that blanched carrots display significantly higher β-carotene content, total carotenoid content, and antioxidant activity in comparison to unblanched carrots [7].

High-pressure processing (HPP) is a cutting-edge non-thermal sterilization method that typically uses pressures between 200 and 600 MPa for food sterilization, enzyme inactivation, and the preservation of food quality attributes such as nutrients, flavour, and texture [8]. A multitude of studies have investigated the use of HPP in carrot juice processing. One study discovered no microbial growth in carrot juice treated at 250 MPa for 15 min (25 °C) [9]. Soysal, Söylemez, and Bozoğlu [10] observed that combining 600 MPa pressure treatment with mild heat at 45 °C was more effective in inactivating peroxidase in carrot juice than heat treatment at 75 °C. Patterson, McKay, Connolly, and Linton [11] reported that HPP of carrot juice at 500 or 600 MPa for 1 min lowered the total viable count (TVC) from 5.8 log CFU/mL to 1.7 log CFU/mL, whilst maintaining a TVC below 3 log CFU/mL during 22 days of storage at 4 °C. Zhang et al. [3] compared high-pressure treatment (550 MPa, 6 min, 25 °C) with a high-temperature short-time treatment (110 °C, 8.6 s) of carrot juice, finding that both techniques reduced TVC by 4–5 log CFU/mL. However, the HP-treated juice demonstrated higher nutritional content and improved physicochemical and sensory properties. Notably, most prior research has concentrated on comparing high pressure and conventional thermal sterilization’s effects on carrot juice’s microbial and nutritional elements. There are relatively few studies examining the impact of blanching pretreatment, high pressure, and the combination of blanching and high pressure on juice quality.

This study aims to assess the effects of blanching pretreatment, HPP, and their combined application on carrot juice quality. To achieve this, carrots were subjected to blanching (heating at 85 °C for 60 s), HPP (600 MPa for 3 min, 20 °C), and a combination of blanching and HPP. Following these processes, the carrot juice was stored at 4 °C for 15 days. Throughout the storage period, the study monitored changes in microbiological and physicochemical quality, colour, and antioxidant properties.

## 2. Materials and Methods

### 2.1. Carrot Sample Preparation and Treatment Conditions

Ten kilograms of carrots in total (*Daucus carota* L.) were sourced from a traditional market in Kaohsiung City. After being washed with tap water, the carrots were peeled and cut into pieces (about 3.5 cm in diameter and 0.5 cm thick). The chopped samples were subsequently divided into two primary batches. The first batch was juiced without blanching, whilst the second batch underwent blanching in hot water at 85 °C for 60 s before juicing. To produce the carrot juice, the carrot pieces were combined with deionized water in a 1:3 ratio and homogenized using a juicer (Galaxie 8, Osterizer, Brampton, ON, Canada). The homogenized liquid was filtered through a double-layer fine cloth, and the resulting filtrate was collected in 250 mL polyethylene terephthalate (PET) plastic bottles. The first batch of carrot juice was split into a control group (unblanched and non-pressurized) and a high-pressure treatment group. The second batch of carrot juice was divided into a blanching group and a combined blanching and HPP group. All the sample groups were stored at 4 °C for 15 days. Each one of the experiment groups was made in three independent bottles for each storage time. Every three days, the carrot juice samples were analysed for microbial and physicochemical quality, colour, and antioxidant properties. At each sampling point, triplicate analyses were carried out for each group.

### 2.2. High-Pressure Processing

The samples were placed in a high-pressure apparatus (HPP600MPa-6L, Bao Tou KeFa, Baotou, China) featuring a pressure chamber with a 20 cm diameter, a 20 cm depth, and a 6.2 L capacity (Figure 1). The 250 mL PET bottles containing the carrot juice were put into the porous plastic basket, and then the basket was put into the high-pressure chamber. This high-pressure device uses high hydrostatic pressure, that is, water as the medium for pressure transmission. To generate high pressure, water is pumped from the reservoir tank into the chamber increasing the pressure applied, whereas water is pumped from the chamber into the reservoir tank to decrease the pressure. The working pressure range spanned from 0.1 to 600 MPa. A water solution was utilized as the pressure transmission medium at 20 ± 2 °C. The average temperature increase in the pressurized fluid was 2 ± 0.5 °C for every 100 MPa increment in pressure. The maximum pressure was attained within 1.5 min, whilst depressurization took about 10–15 s. The HPP parameters for the samples were set at 600 MPa for a 3 min duration.

### 2.3. Microbial Analysis

For the total plate count (TPC) of live bacteria, the juice samples were taken aseptically into a vertical laminar-flow cabinet (#3970420, Labconco Corporation, Kansas City, MO, USA) and 1 mL of carrot juice was extracted under sterile conditions and combined with 9 mL of saline solution in a sterilized test tube, creating 10-fold serial dilutions. From the pure juice and the various dilutions, 0.1 mL (in duplicate) was spread onto trypticase soy agar (TSA, Difco, BD, Sparks, MD, USA) culture medium. The plates were incubated at 35 °C for 48 h, after which the number of growing colonies was determined and expressed as log colony forming unit (CFU) per millilitre. The TPC measurement of each sample was performed in duplicate.

To detect coliforms and *Escherichia coli*, the three-tube most probable number (MPN) method was employed, as outlined in the *Bacteriological Analytical Manual* of the United States Food and Drug Administration [12]. Briefly, 1.0 mL of the 10-fold serial dilutions prepared from the abovementioned TPC detection was transferred to Lauryl Sulfate Tryptose (LST) broth (Difco, BD, Sparks, MD, USA) and a Brilliant Green Lactose Bile Broth (BGLB) (Difco, BD, Sparks, MD, USA) tube. After incubation at 35 °C for 48 h, gas production in LST and BGLB tubes was considered the positive reaction, and the MPN of coliform was determined. Furthermore, one loop of the solution taken from the gas-producing BGLB tube was transferred to *E. coli* broth (EC, Difco, BD, Sparks, MD, USA)) and incubated in 45 °C for 24 h. If there was gas production in the EC tube, one loop was streaked onto eosin methylene blue agar and incubated in 35 °C for 24 h. Colonies with a metallic lustre were confirmed as *E. coli*, and the MPN of *E. coli* was determined. The coliforms and *E. coli* measurements of each sample were performed in duplicate. Additionally, CHROMagar™ Salmonella Plus (CHROMagar, Paris, France) was used to assess the presence of *Salmonella* spp. The handling and incubation procedures for CHROMagar were carried out per the manufacturer’s guidelines. The *Salmonella* spp. measurement of each sample was performed in duplicate.

### 2.4. Chemical Quality Analysis

For pH measurements, a pH meter (PL-700 PV, Mettler Toledo, Hong Kong, China) was initially calibrated using standard buffer solutions with pH values of 4.01, 6.86, and 9.18. Subsequently, the pH of the carrot juice samples was directly determined. The pH measurement of each sample was performed in duplicate. To assess total soluble solids (TSS), the carrot juice samples underwent centrifugation (8000× *g*, 15 min) (Heraeus Megafuge 8R Centrifuge, Thermo Fisher Scientific, Waltham, MA, USA), and the resulting supernatant was analysed with a WAY-2S digital Abbe Refractometer (Model RMT K 7121, Chuanhua Prescision Co., Taipei, Taiwan). TSS values were expressed in degrees Brix. The TSS measurement of each sample was performed in duplicate. Regarding total titratable acidity (TTA), a method adapted from Lisiewska and Kmiecik [13] was utilised. A 0.02 M NaOH standard solution titrated 10 g of the carrot juice sample, with phenolphthalein serving as the indicator to identify the titration endpoint at a pH of 8.2 ± 0.1. TTA values were presented as citric acid equivalents. The TTA measurement of each sample was performed in duplicate.

### 2.5. Colour Analysis

The colour characteristics of the juice samples, encompassing Hunter *L* (lightness), *a* (redness), and *b* (yellowness), were assessed utilizing an SA-2000 NIPPON DENSHOKU colorimeter (Osaka, Japan). For the evaluation of browning degree (BD), carrot juice samples were subjected to centrifugation at 4 °C with an 8000× *g* force for 8 min (Heraeus Megafuge 8R Centrifuge). The supernatant was subsequently filtered through a 0.45 μm nitrocellulose membrane. The filtrate’s absorbance was recorded at a 420 nm wavelength using a U-1800 Spectrophotometer (Hitachi, Tokyo, Japan), following the methodology outlined by Wang et al. [14]. The colour measurement of each sample was performed in duplicate.

### 2.6. Antioxidant Capacity Analysis

To evaluate the total phenolic content, carrot juice samples were centrifuged (8000× *g*, 4 °C, 15 min) (Heraeus Megafuge 8R Centrifuge), and the supernatant was subjected to a reaction with the Folin–Ciocalteu reagent, following the method outlined by Aadil et al. [15]. After the reaction, the sample absorbance was measured at 760 nm using a U-1800 Spectrophotometer (Hitachi, Tokyo, Japan). The results were presented as milligrams of gallic acid equivalents per 100 g of carrot juice. The total phenolic measurement of each sample was performed in duplicate. Regarding DPPH radical scavenging activity, a method adapted from Brand-Williams et al. [16] was employed. Overall, 1 mL of the centrifuged juice supernatant was combined with 2 mL of DPPH (1,1-diphenyl-2-picrylhydrazyl) solution (30 μM) and left to react in the dark for 30 min. Subsequently, the absorbance was recorded at 517 nm using a U-1800 Spectrophotometer. The DPPH radical scavenging activity measurement of each sample was performed in duplicate. For the assessment of β-carotene content, 0.6 g of the juice sample was mixed with 6 mL of butylated hydroxytoluene ethanol solution (1 g/100 mL) for 20 s. The mixture was heated in an 80 °C water bath (SB-10, Jaan-Yuh, Taipei, Taiwan) for 5 min, removed, and then combined with 0.5 mL of 80% KOH before being heated in an 85 °C water bath for 10 min. The mixture was then promptly cooled on crushed ice. The cooled solution was blended with 3 mL of deionised water and 3 mL of hexane, followed by centrifugation (Heraeus Megafuge 8R Centrifuge) to collect the upper layer. The residue underwent two additional extractions with hexane, and the combined upper layers were adjusted to a volume of 10 mL with hexane. The absorbance values at 450 nm (A_450_) and 503 nm (A_503_) were measured using a spectrophotometer (U-1800), and the β-carotene content was calculated based on the formula provided by Sanusi and Adebiyi [17]. The β-carotene measurement of each sample was performed in duplicate.
β-carotene content = 4.642 × A_450_ − 3.091 × A_503_(1)

### 2.7. Statistical Analysis

For this study, SPSS version 12.0 software (St. Armonk, NY, USA) was employed to examine the differences amongst distinct treatment groups. All values were reported as the mean ± standard deviation, derived from triplicate measurements. Statistical evaluations were carried out using both the Tukey test and one-way ANOVA, with a *p*-value below 0.05 signifying a statistically significant discrepancy.

## 3. Results and Discussion

### 3.1. Impact of Blanching and HPP on the Microbiological Quality of Carrot Juice

This study categorized carrot juice into several groups: control (untreated), blanching, HPP, and a combination of blanching and HPP (Blanching/HPP). The microbiological quality outcomes after 15 days of storage at 4 °C are displayed in Table 1 and Figure 2, Figure 3 and Figure 4. As illustrated in Table 1, the TPC in the control group diminished from 4.6 log CFU/mL to 3.38 log CFU/mL (*p* < 0.05) after blanching. In contrast, no TPC was detected in either the HPP or blanching/HPP samples. Moreover, coliform counts of 3.0 MPN/mL were identified in both the control and blanching groups, whereas the HPP and blanching/HPP samples showed no presence of coliform (Table 1). Furthermore, all groups exhibited no detection of *E. coli*, but the control group revealed *Salmonella* spp. at 2.30 log CFU/mL, which was absent in the other groups (Table 1). Based on these observations, it can be inferred that blanching marginally reduced TPC by 1.22 log CFU/mL but was incapable of completely eliminating coliform. Conversely, HPP (600 MPa, 3 min, 20 °C), irrespective of blanching pretreatment, effectively eradicated TPC, coliform, and *Salmonella* spp. from the carrot juice. These findings are consistent with a study by Zhang et al. [3], who reported that blanched carrot juice subjected to HPP (550 MPa, 6 min, 25 °C) experienced a TPC reduction of approximately 4.30 log CFU/mL. Patterson et al. [11] also presented similar results, demonstrating that unblanched carrot juice exposed to HPP (500 or 600 MPa, pressurized for 1 min, 25 °C) saw a decrease in TPC from 5.8 log CFU/mL to 1.7 log CFU/mL.

The TPC of the control group consistently increased throughout the 15-day storage period at 4 °C (Figure 2). The TPC of the blanching group displayed a gradual rising trend during the first nine days of storage, after which it rapidly increased, showing no difference compared to the control group (*p* > 0.05). However, no TPC was detected for both the HPP and blanching/HPP samples throughout the 15-day storage period (Figure 2).

Similarly, the changes in coliform and *Salmonella* spp. counts followed the pattern of TPC. The coliform and *Salmonella* spp. counts in the control group steadily increased with storage duration. In contrast, the blanching group maintained stable coliform counts and no growth of *Salmonella* spp. during the first 6 days of storage, but both counts rapidly increased after day 9 (Figure 3 and Figure 4). Moreover, neither coliform nor *Salmonella* spp. were detected in the HPP and blanching/HPP samples during the 15-day storage period (Figure 3 and Figure 4). According to the microbial hygiene standards for fruit juice beverages set by the Taiwan Food and Drug Administration, TPC must be below 10^4^ CFU/mL, coliform below 10 MPN/mL, and no detection of *E. coli* and *Salmonella* spp. is allowed. The present study revealed that the TPC and *Salmonella* spp. of freshly squeezed carrot juice (control) exceeded the standard of 10^4^ CFU/mL and the no-detection requirement, respectively, whilst coliform surpassed the hygiene standard (10 MPN/mL) on the 3 days of storage. For samples treated with blanching, TPC, coliform, and *Salmonella* spp. exceeded the hygiene standard on day 9, indicating a refrigerated shelf life of only 6 days for blanched samples. This study demonstrated that HPP (600 MPa, 3 min, 20 °C), regardless of blanching pretreatment, effectively inhibited microbial growth in samples during the refrigerated storage period (15 days), thus extending their shelf life to 15 days. These findings concur with those of Patterson et al. [11], who found that TPC remained below 3 log CFU/mL during a 22-day storage period at 4 °C for unblanched carrot juice treated with HPP (500 or 600 MPa, 1 min, 25 °C). Similarly, Zhang et al. [3] reported that for blanched carrot juice subjected to HPP (550 MPa, 6 min, 25 °C), TPC remained below 2.0 log CFU/mL throughout the refrigerated storage period.

### 3.2. Impact of Blanching and HPP on the Chemical Quality of Carrot Juice

Table 2 illustrates the changes in pH, TSS, and TTA for carrot juice samples subjected to various treatments during refrigerated storage. On day zero, the pH values of the blanching and blanching/HPP samples were lower compared to the control and HPP groups, suggesting that blanching leads to a decrease in carrot juice pH. This is likely due to heat treatment causing cell disruption or particle dissolution in carrots, which releases organic acids and subsequently lowers the pH [18]. However, during the middle stages of storage, the pH values of the control group (from day 6) and blanching group (from day 9) declined over time, whilst the pH values of the HPP and blanching + HPP groups remained relatively stable (Table 2). This may be attributed to significant microbial growth in untreated and blanched carrot juice during storage, resulting in acid production, particularly when lactic acid bacteria become dominant and ferment to produce lactic acid, causing juice spoilage [11].

A marked decrease (*p* < 0.05) was observed in the control and blanching groups’ TSS values during the later stages of storage, whilst the TSS values of the HPP and blanching/HPP samples exhibited minimal change (Table 2). This could be due to substantial microbial growth in untreated and blanched carrot juice during the later stages of storage, which concurrently metabolises or consumes sugars in the juice, leading to a reduction in TSS [3]. Conversely, an increase (*p* < 0.05) was noted in the control and blanching groups’ TTA values in the later stages of storage (from day 9), whereas the TTA values of the HPP and blanching/HPP samples remained relatively stable (Table 2). Patterson et al. [11] highlighted that lactic acid bacteria become one of the dominant species in untreated carrot juice during refrigerated storage. As such, the increase in TTA in the untreated and blanched groups in this study could be due to extensive microbial growth and fermentation, particularly by lactic acid bacteria, which produce lactic acid during storage. 

### 3.3. Impact of Blanching and HPP on the Colour of Carrot Juice

Table 3 demonstrates the alterations in colour values and BD for carrot juice samples exposed to different treatments during refrigerated storage. It is apparent that from the midpoint of the storage period, the *L* values (day 6), *a* values (day 9), and *b* values (day 9) of the control and blanching groups declined over time (*p* < 0.05), whilst their BD markedly increased after day 12. In contrast, for the HPP and blanching/HPP samples, apart from a minor decrease in *a* values during the later stages of storage (after day 9), other values such as *L* values, *b* values, and BD remained relatively stable throughout the refrigerated storage period (Table 3). Moreover, the BD values of the HPP group were consistently lower than those of the other treatment groups during storage, indicating the least degree of browning. In summary, the present study reveals that the colour of untreated (control group) and blanched carrot juice darkens and becomes browner during the later stages of storage. This may be due to substantial microbial growth in untreated and blanched carrot juice during the later stages of storage, causing discolouration from an increase in bacterial biomass, or potentially from browning generated by the Maillard reaction and the associated breakdown of pigments [14]. Conversely, carrot juice subjected to HPP (with or without blanching) exhibited minimal colour changes, indicating that HPP can effectively delay alterations in colour, and, in particular, the browning of carrot juice. Zhang et al. [3] reported similar observations, noting that high-pressure-treated (550 MPa, 6 min, 25 °C) carrot juice samples showed no significant changes in *L* values, *a* values, *b* values, or BD values during 20 days of refrigerated storage.

### 3.4. Impact of Blanching and HPP on the Antioxidant Properties of Carrot Juice

Table 4 illustrates the variations in total phenolic content, DPPH radical scavenging capacity, and β-carotene levels in carrot juice samples exposed to different treatments during refrigerated storage. Concerning total phenolic content, it is clear that on day zero, the blanching group exhibited a lower total phenolic content (0.94 mg/100 mL) compared to the other groups. However, as storage time progressed, the total phenolic content increased, reaching its peak (6.71 mg/100 mL) on the final day, which was higher than the other groups (*p* < 0.05). Moreover, the total phenolic content in the control group rose with storage time, attaining its maximum on day 9 (7.53 mg/100 mL) before gradually declining. The findings of this study suggest that the total phenolic content of blanched carrot juice is reduced, potentially due to the oxidation or degradation of phenolic compounds during the heating process, as intense thermal treatment can cause a decrease in phenolic content [19,20]. In addition, during storage, the carrot juice from the control and blanching groups underwent cellular structural ageing and degradation changes, leading to the release of free phenolic acids and free amino acids, which in turn increased the total phenolic content [21].

Furthermore, the HPP group displayed minimal changes in the total phenolic content during storage, suggesting that HPP does not significantly affect the total phenol concentrations in carrot juice. Similarly, Landl et al. [20] found that HPP at 400 MPa had no appreciable impact on apple juice’s total phenolic content. Likewise, Barba et al. [22] observed no marked differences in total phenolic content in vegetable juices after HPP (100–400 MPa/9 min, 25 °C). However, Zhang et al. [3] noted a slight decline in total phenol levels in carrot juice exposed to HPP (550 MPa, 6 min, 25 °C) as the storage time increased. This could be due to the oxidation and degradation of phenolic compounds during storage caused by high pressure, as well as the aggregation of phenolic compounds with proteins [3,23]. Interestingly, some studies have reported a significant increase in phenolic compound levels in orange juice treated with high pressure [24], possibly because high pressure disrupts plant cell structures, making phenolic compounds more readily extractable. This study observed no significant changes in the total phenolic content of carrot juice treated solely with HPP, which may be attributed to a balance being achieved between the enhanced extraction rate of phenolic compounds and their oxidative degradation.

In terms of antioxidant capacity for scavenging DPPH free radicals, the blanching and blanching/HPP samples exhibited greater DPPH-scavenging capabilities on day zero compared to other groups (*p* < 0.05), maintaining a stable level throughout refrigerated storage (Table 4). However, the control group’s DPPH-scavenging capacity increased as the storage time elapsed. Studies have indicated that DPPH free radical scavenging activity rises with increasing phenolic content; but, upon reaching a certain threshold, this capacity begins to wane [25]. Furthermore, blanching may cause alterations in plant tissues, such as cell disruption, pectin breakdown, and cellulose structure loosening, thereby enhancing the extraction of antioxidant substances [26]. Sila et al. [27] also highlighted that heat treatment can lead to pectin degradation, which in turn results in plant tissue softening or damage and an increased release of antioxidant compounds. Therefore, the elevated antioxidant capacity observed in carrot juice treated with blanching (irrespective of HPP) in this study could be linked to pectin decomposition prompted by heat treatment, causing plant tissue softening or damage and augmenting the release of antioxidant compounds. Additionally, the HPP samples displayed minimal change in antioxidant capacity during storage, signifying that HPP alone does not have a substantial impact on carrot juice’s antioxidant capacity (Table 4). Conversely, Barba et al. [28] found that the antioxidant capacity of blueberry juice treated at >400 MPa was inferior to that of untreated juice. Moreover, Zhang et al. [3] observed a declining DPPH scavenging ability in carrot juice treated with HPP (550 MPa, 6 min, 25 °C) as storage time progressed, which could be associated with a decrease in antioxidant substances such as total phenol. Thus, this study revealed that the single HPP group maintained consistent levels of antioxidant substances and total phenol content during storage, leading to unchanging antioxidant activity.

Concerning β-carotene content, this study found that all groups experienced a gradual increase in β-carotene levels as storage time advanced (Table 4). On the 15th day, the HPP samples exhibited marginally higher β-carotene content (0.764 µg/mL) compared to other groups. One plausible explanation for this observation is that high pressure may induce the denaturation of protein–carotenoid complexes, allowing carotenoids to be slowly released during storage and consequently increasing the number of extractable carotenoids [28]. The outcomes of this study align with those of Carbonell-Capella et al. [29], who found that pressures above 300 MPa could notably boost the carotenoid content in fruit juices. Furthermore, other research has demonstrated that HPP (350 MPa/30 °C/5 min) significantly elevates the carotenoid content in orange juice.

In the study of Patterson et al. [11], carrots were cut into pieces and directly squeezed for juice, and then subjected to two HPP samples (500 MPa and 600 MPa, 1 min, 25 °C), to evaluate the TPC changes in HPP samples during low-temperature storage. However, in Zhang et al.’s [3] study, the diced carrots were first blanched (90 °C, 2 min) and then squeezed, and then divided into an HPP group (550 MPa, 6 min, 25 °C) and a high-temperature short-time (HTST) (110 °C, 8.6 s) group to evaluate the changes in the chemical and antioxidant properties of both samples during cold storage. Therefore, the novelty of this study is to integrate the above two studies, that is, to divide carrot juice into blanching, HPP, and blanching combined with HPP groups, and to evaluate the changes in the microbial and physicochemical quality, colour, and antioxidant properties of different treatment groups stored at 4 °C.

Compared with traditional thermal sterilization processing, HPP, a non-thermal processing technology, can retain the natural nutrients, flavour, and colour of carrot juice [30,31]. At the same time, HP-treated juice is also regarded as a "clean label food" because of a lack of preservative and food additive addition [30,31]. Recently, due to people’s demands for naturalness, nutrition, and health, high-pressure juice has been commercially developed, and the world’s output value has exceeded 10 billion US dollars [30]. In addition, in response to the increasing demand for personalization and health, the supply of high-pressure juice products with a small package volume of 250–350 mL is gradually increasing; moreover, in order to emphasize that high-pressure juice can be consumed in a fresh state, some manufacturers have suggested that the juice should be refrigerated to enjoy the taste, from 2 weeks to 1 month [30]. Generally, the standard treatment method for commercially available high-pressure juice is 600 MPa for 3 min, and this condition has been commonly used in commercial HPP juice products [32].

Packaging materials for HPP products must be capable of withstanding operating pressures, adequately heat-sealable, and elastic to facilitate pressure transmission. Therefore, rigid packaging materials made of metal and glass are not suitable for HPP because they are prone to deformation and cracking under high pressure. Currently, soft polymeric bags, cans, trays, and bottles made of PET, polyethylene (PE), polypropylene (PP), and ethylene vinyl alcohol polymer (EVOH) are commonly used packaging materials for HPP foods [30]. In addition, because high pressure cannot inactivate bacterial spores, if the storage temperature is too high, the spores will germinate and grow, resulting in the spoilage of food. Therefore, high-pressure food is usually recommended to be stored in a refrigerated state [31,33].

## 4. Conclusions

This study demonstrated that treating carrot juice with HPP and blanching/HPP significantly reduced TPC, coliform, and *Salmonella* spp. bacterial counts whilst effectively inhibiting microbial growth throughout refrigerated storage. Based on the microbial limit standards for fruit juice beverages, HPP (600 MPa, 3 min, 20 °C) can prolong the shelf life of carrot juice from the zero day (freshly squeezed juice) and 6 days (blanching-only treatment group) up to 15 days, irrespective of blanching pretreatment. Moreover, HPP (with or without blanching) successfully delays alterations in juice pH, total soluble solids, total titratable acidity, and colour values, particularly in preventing sample browning. In summary, HPP improves the food safety of carrot juice whilst preserving product quality and extending the samples’ shelf life.

## Figures and Tables

**Figure 1 foods-12-03107-f001:**
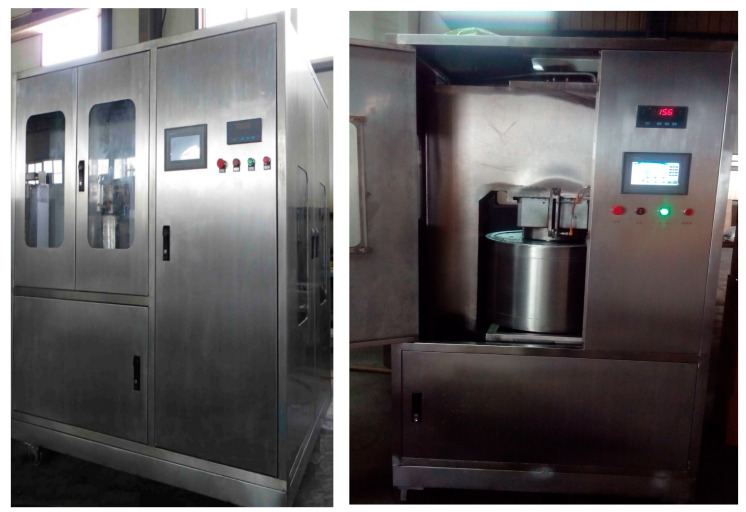
Appearance (**left**) and internal cavity structure (**right**) of high-pressure equipment (Bao Tou KeFa, Baotou, China).

**Figure 2 foods-12-03107-f002:**
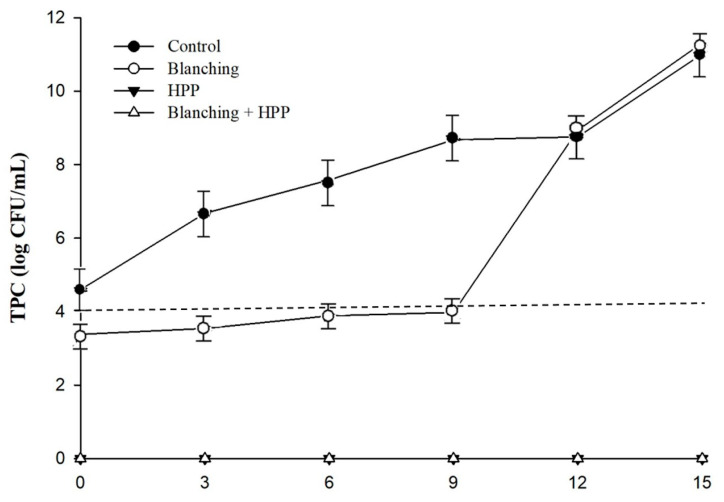
Changes in total plate count (TPC) in untreated (control), blanching, HPP (600 Mpa for 3 min, 20 °C), and blanching combined with HP-treated carrot juice during 15 days of storage at 4 °C. Dashed line represents 4 log CFU/mL of TPC as the limit standard for juice beverage.

**Figure 3 foods-12-03107-f003:**
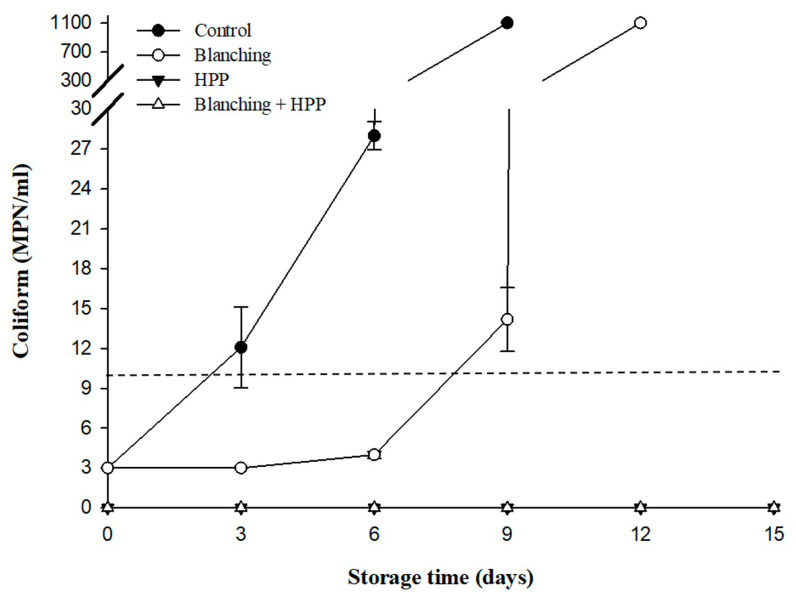
Changes in coliform (MPN/mL) in untreated (control), blanching, HPP (600 MPa for 3 min, 20 °C), and blanching combined with HP-treated carrot juice during 15 days of storage at 4 °C. Dashed line represents 10 MPN/mL of coliform as the limit standard for juice beverage.

**Figure 4 foods-12-03107-f004:**
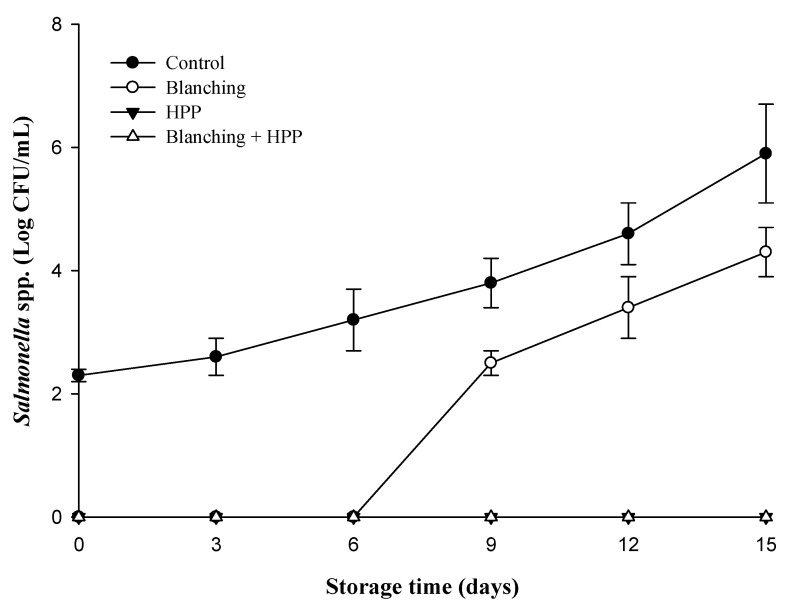
Changes in *Salmonella* spp. in untreated (control), blanching, HPP (600 MPa for 3 min, 20 °C) and blanching combined with HP-treated carrot juice during 15 days of storage at 4 °C.

**Table 1 foods-12-03107-t001:** Total plate count (TPC), coliform, *E. coli* and *Salmonella* spp. in untreated (control), blanching, HPP (600 MPa for 3 min, 20 °C), and blanching combined with HP-treated carrot juice.

Treatments	TPC (log CFU/mL)	Coliform (MPN/mL)	*E. coli* (MPN/mL)	*Salmonella* spp. (log CFU/mL)
Control	4.60 ± 0.03 ^A^*	3.0	<3.0	2.30 ± 0.10
Blanching	3.38 ± 0.05 ^B^	3.0	<3.0	<1.0
HPP	<1.0 ^C^	<3.0	<3.0	<1.0
Blanching + HPP	<1.0 ^C^	<3.0	<3.0	<1.0

* All data were the means ± standard deviation of three replicates (n = 3). ^A–C^: different letters in the same storage time indicate significant differences (*p* < 0.05).

**Table 2 foods-12-03107-t002:** Changes in pH, total soluble solids (TSS), and total titratable acidity (TTA) in untreated (control), blanching, HPP (600 MPa for 3 min, 20 °C), and blanching combined with HP-treated carrot juice during 15 days of storage at 4 °C.

Quality Attributes	Treatments	Storage Time (Days)					
		0	3	6	9	12	15
pH	Control	6.15 ± 0.10 ^aA^*	6.23 ± 0.10 ^aA^	5.91 ± 0.10 ^cB^	5.82 ± 0.10 ^bC^	4.37 ± 0.10 ^dD^	4.27 ± 0.10 ^dE^
	Blanching	5.84 ± 0.10 ^cB^	5.87 ± 0.10 ^cB^	6.06 ± 0.10 ^bA^	6.00 ± 0.13 ^aA^	5.60 ± 0.07 ^cC^	5.42 ± 0.10 ^cD^
	HPP	6.00 ± 0.10 ^bC^	6.12 ± 0.10 ^bB^	6.23 ± 0.05 ^aA^	6.15 ± 0.15 ^aAB^	6.17 ± 0.18 ^aAB^	5.81 ± 0.10 ^bC^
	Blanching + HPP	5.59 ± 0.10 ^dD^	5.76 ± 0.10 ^dC^	5.94 ± 0.08 ^cAB^	5.90 ± 0.10 ^bAB^	5.84 ± 0.25 ^bBC^	5.98 ± 0.20 ^aA^
TSS (°Brix)	Control	2.03 ± 0.10 ^aA^	2.00 ± 0.10 ^aA^	1.97 ± 0.10 ^aAB^	1.83 ± 0.10 ^aB^	1.23 ± 0.10 ^bC^	1.23 ± 0.10 ^bC^
	Blanching	2.07 ± 0.12 ^aA^	1.93 ± 0.10 ^abA^	1.97 ± 0.10 ^aA^	1.93 ± 0.10 ^aA^	1.57 ± 0.10 ^aB^	1.13 ± 0.10 ^bC^
	HPP	1.83 ± 0.10 ^bA^	1.88 ± 0.10 ^bcA^	1.93 ± 0.10 ^aA^	1.83 ± 0.10 ^aA^	1.63 ± 0.10 ^aB^	1.60 ± 0.10 ^aB^
	Blanching + HPP	1.73 ± 0.10 b ^AB^	1.82 ± 0.10 ^cAB^	1.90 ± 0.10 ^aA^	1.90 ± 0.10 ^aA^	1.63 ± 0.10 ^aB^	1.63 ± 0.10 ^aB^
TTA (%)	Control	0.027 ± 0.001 ^bE^	0.025 ± 0.006 ^bE^	0.043 ± 0.003 ^aD^	0.058 ± 0.001 ^aC^	0.150 ± 0.011 ^aB^	0.162 ± 0.002 ^aA^
	Blanching	0.027 ± 0.001 ^bC^	0.032 ± 0.002 ^aC^	0.032 ± 0.003 ^aC^	0.040 ± 0.002 ^bB^	0.042 ± 0.001 ^bB^	0.084 ± 0.001 ^bA^
	HPP	0.032 ± 0.001 ^aC^	0.037 ± 0.002 ^aB^	0.041 ± 0.003 ^aA^	0.040 ± 0.002 ^bAB^	0.038 ± 0.002 ^bB^	0.039 ± 0.001 ^cAB^
	Blanching + HPP	0.028 ± 0.002 ^bC^	0.037 ± 0.002 ^aB^	0.046 ± 0.003 ^aA^	0.041 ± 0.001 ^bAB^	0.040 ± 0.001 ^bB^	0.038 ± 0.001 ^cB^

* All data were the means ± standard deviation of three replicates (n = 3). ^a–d^: different letters in the same column indicate significant differences (*p* < 0.05); ^A–E^: different letters in the same row indicate significant differences (*p* < 0.05).

**Table 3 foods-12-03107-t003:** Changes in colour and browning degree in untreated (control), blanching, HPP (600 MPa for 3 min, 20 °C), and blanching combined with HP-treated carrot juice during 15 days of storage at 4 °C.

Quality Attributes	Treatment	Storage Time (Days)					
		0	3	6	9	12	15
Hunter colour	Control	15.49 ± 0.40 ^bA^*	15.22 ± 0.20 ^aA^	11.70 ± 0.27 ^bB^	11.53 ± 0.07 ^cB^	10.04 ± 0.08 ^dC^	10.23 ± 0.18 ^cC^
*L*	Blanching	16.87 ± 0.07 ^aA^	14.12 ± 0.12 ^bB^	12.66 ± 1.02 ^bC^	12.72 ± 0.11 ^bC^	11.30 ± 0.13 ^cCD^	9.95 ± 0.22 ^cD^
	HPP	14.70 ± 0.14 ^cA^	13.80 ± 0.11 ^bBC^	13.06 ± 0.22 ^bD^	13.43 ± 0.32 ^bCD^	13.19 ± 0.36 ^bD^	14.42 ± 0.06 ^aAB^
	Blanching + HPP	14.35 ± 0.35 ^cBC^	15.17 ± 0.13 ^aAB^	15.99 ± 0.23 ^aA^	15.08 ± 0.59 ^aB^	13.68 ± 0.05 ^aC^	13.86 ± 0.27 ^bC^
*a*	Control	5.42 ± 0.71 ^bAB^	4.62 ± 0.07 ^bB^	5.85 ± 0.13 ^aA^	2.11 ± 0.13 ^dC^	0.64 ± 0.11 ^dD^	0.57 ± 0.31 ^cD^
	Blanching	4.77 ± 0.11 ^bA^	3.75 ± 0.03 ^cB^	4.78 ± 0.14 ^cA^	2.87 ± 0.11 ^cC^	1.64 ± 0.12 ^cD^	0.51 ± 0.02 ^cE^
	HPP	6.22 ± 0.04 ^aA^	5.92 ± 0.02 ^aB^	5.62 ± 0.02 ^aC^	4.13 ± 0.13 ^bE^	3.40 ± 0.11 ^bF^	4.53 ± 0.18 ^aD^
	Blanching + HPP	6.41 ± 0.15 ^aA^	5.80 ± 0.14 ^aB^	5.20 ± 0.15 ^bC^	4.56 ± 0.15 ^aD^	4.21 ± 0.03 ^aDE^	3.86 ± 0.14 ^bE^
*b*	Control	6.8 ± 0.22 ^bB^	7.73 ± 0.26 ^aA^	6.92 ± 0.09 ^aA^	4.02 ± 0.66 ^cB^	3.42 ± 0.34 ^cBC^	2.72 ± 0.18 ^cC^
	Blanching	5.6 ± 0.09 ^cB^	7.19 ± 0.04 ^bA^	6.95 ± 0.31 ^aA^	5.22 ± 0.21 ^bB^	4.57 ± 0.03 ^bC^	2.07 ± 0.32 ^dD^
	HPP	8.56 ± 0.13 ^aA^	7.6 ± 0.13 ^aB^	7.04 ± 0.62 ^aBC^	6.35 ± 0.07 ^aC^	6.40 ± 0.17 ^aC^	7.06 ± 0.10 ^aBC^
	Blanching + HPP	8.44 ± 0.11 ^aA^	7.26 ± 0.21 ^bB^	5.97 ± 0.05 ^bD^	6.67 ± 0.36 ^aBC^	6.23 ± 0.35 ^aC^	5.96 ± 0.28 ^bD^
Browning	Control	0.25 ± 0.01 ^aC^	0.27 ± 0.03 ^aC^	0.14 ± 0.01 ^cE^	0.19 ± 0.01 ^cD^	0.76 ± 0.01 ^aB^	1.20 ± 0.02 ^aA^
degree	Blanching	0.27 ± 0.01 ^aCD^	0.17 ± 0.01 ^bE^	0.22 ± 0.01 ^aD^	0.29 ± 0.01 ^aC^	0.37 ± 0.03 ^bB^	0.42 ± 0.01 ^bA^
	HPP	0.08 ± 0.01 ^cB^	0.08 ± 0.01 ^cBC^	0.08 ± 0.01 ^dC^	0.07 ± 0.01 ^dD^	0.09 ± 0.02 ^dA^	0.08 ± 0.01 ^dBC^
	Blanching + HPP	0.12 ± 0.01 ^bD^	0.15 ± 0.01 ^bC^	0.18 ± 0.01 ^bB^	0.24 ± 0.01 ^bA^	0.20 ± 0.01 ^cA^	0.16 ± 0.01 ^cBC^

* All data were the means ± standard deviation of three replicates (n = 3). ^a–d^: different letters in the same column indicate significant differences (*p* < 0.05); ^A–F^: different letters in the same row indicate significant differences (*p* < 0.05).

**Table 4 foods-12-03107-t004:** Changes in total phenols, scavenging DPPH radicals, and β-carotene in untreated (control), blanching, HPP (600 MPa for 3 min, 20 °C), and blanching combined with HP-treated carrot juice during 15 days of storage at 4 °C.

Quality Attributes	Treatment	Storage Time (Days)					
		0	3	6	9	12	15
Total phenols (mg/100 mL)	Control	1.37 ± 0.03 ^abE^*	1.83 ± 0.03 ^aD^	5.89 ± 0.29 ^aB^	7.53 ± 0.04 ^aA^	5.61 ± 0.01 ^aB^	4.61 ± 0.07 ^bC^
	Blanching	0.94 ± 0.03 ^cE^	1.23 ± 0.06 ^bE^	2.16 ± 0.21 ^bD^	3.51 ± 0.40 ^bC^	5.13 ± 0.01 ^bB^	6.71 ± 0.06 ^aA^
	HPP	1.63 ± 0.03 ^aD^	1.77 ± 0.03 ^aC^	1.83 ± 0.03 ^cC^	2.23 ± 0.03 ^cA^	1.93 ± 0.01 ^dB^	2.17 ± 0.03 ^cA^
	Blanching + HPP	1.30 ± 0.24 ^bD^	1.34 ± 0.11 ^bD^	2.37 ± 0.03 ^bC^	3.11 ± 0.03 ^bB^	4.24 ± 0.01 ^cA^	4.59 ± 0.04 ^bA^
Scavenging DPPH radicals (%)	Control	17.99 ± 0.88 ^bD^	23.04 ± 0.42 ^cC^	27.05 ± 0.96 ^bA^	27.16 ± 0.55 ^bA^	27.28 ± 0.16 ^bA^	25.05 ± 0.32 ^bB^
	Blanching	29.69 ± 0.42 ^aC^	30.87 ± 0.27 ^aBC^	30.97 ± 0.42 ^aBC^	32.08 ± 0.16 ^aAB^	32.92 ± 0.82 ^aA^	30.46 ± 0.47 ^aC^
	HPP	17.03 ± 0.42 ^bA^	18.44 ± 1.25 ^dA^	18.31 ± 0.99 ^cA^	16.85 ± 1.40 ^cA^	15.94 ± 0.42 ^cA^	15.94 ± 0.42 ^cA^
	Blanching + HPP	28.69 ± 0.95 ^aA^	28.23 ± 0.83 ^bA^	27.78 ± 0.83 ^bA^	27.32 ± 0.71 ^bA^	26.87 ± 0.69 ^bA^	24.00 ± 0.69 ^bB^
β-carotene (μg/mL)	Control	0.551 ± 0.008 ^bC^	0.592 ± 0.029 ^aB^	0.613 ± 0.017 ^abB^	0.667 ± 0.014 ^aA^	0.659 ± 0.004 ^bA^	0.647 ± 0.011 ^cA^
	Blanching	0.563 ± 0.010 ^bC^	0.576 ± 0.002 ^abCD^	0.591 ± 0.003 ^bC^	0.583 ± 0.001 ^cD^	0.611 ± 0.005 ^dB^	0.731 ± 0.010 ^bA^
	HPP	0.605 ± 0.006 ^aD^	0.608 ± 0.009 ^aD^	0.642 ± 0.017 ^abC^	0.644 ± 0.001 ^bC^	0.683 ± 0.008 ^aB^	0.764 ± 0.009 ^aA^
	Blanching + HPP	0.556 ± 0.009 ^bD^	0.538 ± 0.009 ^bD^	0.611 ± 0.010 ^aC^	0.596 ± 0.004 ^cC^	0.642 ± 0.001 ^cB^	0.751 ± 0.005 ^abA^

* All data were the means ± standard deviation of three replicates (n = 3). ^a–d^: different letters in the same column indicate significant differences (*p* < 0.05); ^A–E^: different letters in the same row indicate significant differences (*p* < 0.05).

## Data Availability

The data presented in this study are available on request from the corresponding author. The data are not publicly available due to privacy and ethical reasons.

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
