# Peer review of "Effect of High-Pressure Processing on the Qualities of Carrot Juice during Cold Storage"

_foods, 2023, doi:10.3390/foods12163107_

Round 1
Reviewer 1 Report
This study assesses the effects of blanching pretreatment, HPP, and their combined application on carrot juice quality in terms of microbiological and chemical qualities, colour, and antioxidant properties of carrot juice stored at 4°C for 15 days.
The major part of the manuscript is based on high pressure. The description given in the experimental section is not enough to validate how the sample was placed in the cell and how the pressure in the cell was increased. How was the temperature and pressure monitored? Was hydrostatic pressure used? A figure showing the setup of the high pressure apparatus is highly recommended
The results are .well described. There are minor corrections that can uplift the quality of the manuscript;
section 2.1
in a ratio
Section 2.7
Is 6.2 L capacity correct when comparing the other dimensions of the cell.
2.3
Needs to be more clear about the analysis if the laminar flow fume cupboard was used etc.
Section 2.6
Why is the font of the equation enlarged?
The font size of the legends in the figures need to be enlarged to become more clear.
In the text when 0 days are used, try replacing 0 with the word zero.
Table 4.
The variables on the left column need to be separated well. The last two variables with their corresponding data are confusing to read.
quality of English Grammar is good. Some minor corrections.
Reviewer 2 Report
I believe that the manuscript submitted for review is correct. The introduction is a good background and introduction to the subject of research, the purpose of the work has been clearly and properly defined. The proposed methods have been properly selected and also described accordingly. The results were also presented appropriately using various graphical forms, properly selected.
I have two minor remarks: first, in the methodological part in paragraph 2.3. I don't really understand how the dilutions were made. This presentation in exponential form is not the right choice. Secondly, I also have a suggestion for paragraph 3.3, where the authors presented the results for the color measurement. I would suggest that the differences in the individual color components be presented as a delta, which is a parameter that is customarily adopted to determine the significance of differences between the L a and b components
Reviewer 3 Report
Comments to Hwang et. al.
pH of ~ 6 makes carrot juice a low acid juice and must be treated to prevent outgrowth of C. bot if it is held anaerobically. That’s why carrot juice is treated at 121 C for juice stored anaerobically typically at room temperature. I have trouble understanding why such an expensive processing method like HPP would be used to get two weeks of refrigerated storage?
Before jumping into M&M give the reader an idea of the volume of consumption for carrot juice. For the USA all juices combined the total consumption is 17 L and probably the majority of that are fruit juices—so carrot would be a tiny amount.
There are serious questions with your M&M and how what you did would be applicable to commercial carrot juice processing—please address. Is there commercially processed HPP or thermally processed carrot juice available in the market to use as a bench mark. Obviously shelf-stable carrot juice with be commercially sterile.
The reader is left not knowing what type of packaging was used? Was this stored anaerobically or just refrigerated? Most carrot juice I’m familiar with is all “shelf stable” unless it’s a “juice bar” and is juiced at the time of consumption.
Good detail on the quality control methods.
Provide additional details on Zhang (3) and Patterson (11) how were your treatment different and what contribution to the literature is your research making? Good contrast with their results and yours later on in the Discussion
It is hard to see the HPP symbols on Fig 1
Nice job on Table 2 presentation of the data—remind the reader of the number of reps per Table 4
Reviewer 4 Report
The article titled "Comparative Effect of Blanching and High-Hydrostatic Pressure Treatments on the Microbiological and Chemical Qualities of Carrot Juice During Cold Storage" is well-written and addresses an important topic in food preservation. The study investigates the impact of blanching and high-hydrostatic pressure (HPP) treatments on the microbiological and chemical qualities of carrot juice during cold storage. However, there are several concerns that need to be addressed.
Major Concerns:
Novelty:
The main concern regarding the article is its novelty. The use of HPP for food preservation has been researched for a while, and the references cited in the manuscript are relatively old, with the most recent being from 2017. The authors should update the references with more recent studies to demonstrate the current relevance and advancement of their work in the field.
Minor comments:
- The font size and equation numbering need to be carefully checked for consistency throughout the manuscript. In particular, the "b-carotene" equation should be clearly labeled and appropriately formatted for clarity and readability.
- Table 2 lacks a legend explaining the meaning of superscripts used in the table. The authors should provide a clear and concise explanation of each superscript to ensure the proper understanding of the data presented in the table.
In conclusion, while the article is well-written and explores an important topic in food preservation, there are major concerns regarding its novelty and the outdated references used.
Round 2
Reviewer 1 Report
Most of the comments have been addressed but there is still concern about the methodology of how the sample was placed in the high pressure chamber. The extraction of the juice is well explained. Blanching is well explained. However, when the juice is placed in the high pressure chamber, the manuscript still does not provide sufficient information on how this part was carried out. For example was it placed in a plastic bag, bottle etc? When the pressure was applied to the sample, was the pressure hydrostatic from water on the sample?
These aspects of the experimental section needs to be clarified.
Reviewer 3 Report
Thank you for being responsive to my concerns and addressing them appropriately!
Author Response
Response to Reviewer 3 Comments:
- Thank you for being responsive to my concerns and addressing them appropriately!
Response 1→Thanks for the review’s comment. We appreciate the reviewers’ efforts in providing valuable comments for us to improve the quality of the manuscripts.